# Effects of Dimerization, Dendrimerization, and Chirality in p-BthTX-I Peptide Analogs on the Antibacterial Activity and Enzymatic Inhibition of the SARS-CoV-2 PL^pro^ Protein

**DOI:** 10.3390/pharmaceutics15020436

**Published:** 2023-01-28

**Authors:** Natália Vitória Bitencourt, Gabriela Marinho Righetto, Ilana Lopes Baratella Cunha Camargo, Mariana Ortiz de Godoy, Rafael Victorio Carvalho Guido, Glaucius Oliva, Norival Alves Santos-Filho, Eduardo Maffud Cilli

**Affiliations:** 1Department of Biochemistry and Organic Chemistry, Institute of Chemistry, São Paulo State University (UNESP), Araraquara 14800-060, SP, Brazil; 2São Carlos Institute of Physics, University of São Paulo, São Carlos 13563-120, SP, Brazil

**Keywords:** p-BthTX-I, p-Bth, multidrug-resistant bacteria, antimicrobial peptide, dendrimers, PL^pro^, SARS-CoV-2, COVID-19

## Abstract

Recent studies have shown that the peptide [des-Cys^11^,Lys^12^,Lys^13^-(p-BthTX-I)_2_K] (p-Bth) is a p-BthTX-I analog that shows enhanced antimicrobial activity, stability and hemolytic activity, and is easy to obtain compared to the wild-type sequence. This molecule also inhibits SARS-CoV-2 viral infection in Vero cells, acting on SARS-CoV-2 PL^pro^ enzymatic activity. Thus, the present study aimed to assess the effects of structural modifications to p-Bth, such as dimerization, dendrimerization and chirality, on the antibacterial activity and inhibitory properties of PL^pro^. The results showed that the dimerization or dendrimerization of p-Bth was essential for antibacterial activity, as the monomeric structure led to a total loss of, or significant reduction in, bacterial activities. The dimers and tetramers obtained using branched lysine proved to be prominent compounds with antibacterial activity against Gram-positive and Gram-negative bacteria. In addition, hemolysis rates were below 10% at the corresponding concentrations. Conversely, the inhibitory activity of the PL^pro^ of SARS-CoV-2 was similar in the monomeric, dimeric and tetrameric forms of p-Bth. Our findings indicate the importance of the dimerization and dendrimerization of this important class of antimicrobial peptides, which shows great potential for antimicrobial and antiviral drug-discovery campaigns.

## 1. Introduction

The growing bacterial resistance, combined with the decline in the discovery of new antibiotics, makes the post-antibiotic era, in which minor injuries can result in death, increasingly possible [1,2]. One of the ways to circumvent the situation triggered by bacterial resistance is to invest in research and development focusing on new molecules with antibacterial potential, among which antimicrobial peptides (AMPs) stand out. These molecules are constituents of the defense systems of several organisms, are widely distributed in nature, and act against various pathogens, including bacteria, fungi, *Leishmania*, and viruses [3,4]. AMPs are promising novel antibiotics that kill multidrug-resistant bacteria and show activity against some bacteria that are considered a priority by the WHO [5]. Their sequence normally has from 10 to 50 amino acids and is modulated by structure–function relationships [6]. The charge, amphipathicity, hydrophobicity and helicity of peptides are fundamental to their biological activity [7]. Although promising, AMPs face challenges regarding their therapeutic application, such as poor pharmacokinetic properties, including low absorption and stability, due to their rapid proteolytic degradation, in addition to their high production cost [8]. Strategies to address these issues include structural modifications, such as dendrimerization and chirality modifications [9]. Many studies have shown that the complete replacement of L with D-amino acids in the peptide sequence to obtain D-enantiomers leads to conservation or increased antimicrobial activity, in addition to increased resistance to proteolytic degradation [10].

Dendrimeric antimicrobial peptides (DMPAs) are commonly obtained from poly-L-lysine nuclei, in which lysine units are used as branch points with the growth of identical peptide chains of linear AMPs in their α and ε amino groups [7,11,12]. This strategy leads to the production of peptides with enhanced antimicrobial activity compared to their monomers. The enhanced activity is due to multivalence; that is, a greater number of peptide chains, leading to greater local concentration and enhanced proteolytic stability due to steric hindrance, and thereby decreasing susceptibility to proteases [13].

Recently, a cationic peptide, named p-BthTX-I, was designed based on the sequence between residues 115-129 of the C-terminal region of Bothropstoxin I (BthTX-I). The peptide p-BthTX-I was previously studied by our research group and showed high antibacterial potential, especially in its dimeric form, linked through disulfide bonds between cysteine residues in KKYRYHLKPFCKK (p-BthTX-I)_2_. In the monomeric form, the peptide undergoes oxidation and forms a dimer in solution and culture medium, showing that the dimeric form is responsible for the antibacterial activity of the p-BthTX-I peptide. Moreover, this peptide did not show activity against *Candida albicans* or cytotoxicity against macrophages, erythrocytes, and epithelial cells, indicating promising selective toxicity [14].

The dimeric peptide (p-BthTX-I)_2_ was subjected to stability tests in serum and its stable by-product [des-Lys^12^,Lys^13^-(p-BthTX-I)_2_] showed similar antibacterial potential and, in some cases, was superior to the original molecule [15]. Furthermore, this peptide showed different mechanisms of action against Gram-positive and negative bacteria [16]. To avoid the oxidation step and make the peptide more stable, our group promoted dimerization using a lysine residue as a branch point. The analog [des-Cys^11^,Lys^12^,Lys^13^-(p-BthTX-I)_2_K], of sequence (KKYRYHLKPF)_2_K (p-Bth), was the most potent analog of the series [17].

Antimicrobial peptides were also reported as potential antiviral compounds [18]; thus, we assessed the p-Bth and analogs’ inhibitory activities against SARS-CoV-2 infection in Vero cells. These molecules showed a potent inhibition of viral infection, acting on the inhibition of PL^pro^ from SARS-CoV-2, a key enzyme in the viral replication process and innate immunity inhibition [19]. Moreover, p-Bth demonstrated antithrombotic activity, possibly due to kallikrein inhibition, suggesting its strong biotechnological potential [20].

Thus, this study aimed to assess and compare L- and D-p-Bth analogs (monomer, dimer and tetramer) regarding their antibacterial activity and inhibition of PL^pro^ of SARS-CoV-2.

## 2. Materials and Methods

### 2.1. Peptides Synthesis

Peptides were manually synthesized through solid-phase peptide synthesis (SPPS) using the methodology described by Merrifield, 1963 [21]. To obtain dimers and tetramers, the amino acid Fmoc-Lys(Fmoc)-OH was used at the beginning of the synthesis as a branch point, allowing for the growth of peptide chains from the α-amino and ε-amino groups of lysine, as described by Lorenzón et al., 2012 and Santos-Filho et al., 2021 [7,17]. After synthesis, the peptides were purified by semi-preparative HPLC using a Shimadzu chromatograph (Tokyo, Japan) equipped with a C18 Jupiter column of 25 × 1 cm, with a particle size of 10 μm. The purity contents (<95%) of the obtained materials were determined by analytical HPLC in a Shimadzu chromatograph (Kyoto, Japan) column (0.46 × 15 cm) of reverse-phase C18, with a particle size of 5 μm (Agilent, Santa Clara, CA, USA). Confirmations that the desired materials were obtained were evaluated by mass spectrometry, using an Ion Trap MS mass spectrometer (Bruker), direct injection and positive detection mode.

### 2.2. Determination of the Minimum Inhibitory Concentration and Minimum Bactericidal Concentration

Bacterial isolates from the American Type Culture Collection (ATCC) were obtained from the Collection of Reference Bacteria on Health Surveillance, Oswaldo Cruz Foundation (FIOCRUZ) after phenotypic and molecular quality control and were registered at SISGEN under the number ADB556B. The minimum inhibitory concentration (MIC) was determined as previously described by Santos-Filho et al., 2021 [17], using the broth microdilution method. The MIC was the lowest concentration that inhibited microbial growth. A total of 100 μL from each well, without growth from the MIC assay, was subcultured on CA-MH agar plates and incubated at 37 °C for 24 h to determine the minimum bactericidal concentration (MBC). MBC was defined as the lowest peptide concentration with no visible growth on the plate. An MBC/MIC ratio >4 suggests bacteriostatic activity in a peptide [22]. The assay was performed in triplicate, and the values represent the mode of the obtained results.

### 2.3. Hemolysis Assay

Peptide hemolytic activity was determined as previously described^15^. Peptides were incubated with erythrocytes in the range from 512 μg/mL to 0.06 μg/mL for 1 h at 37 °C. Triton X-100 1% was used as a positive control for hemolysis. Assays were performed in duplicate. The Research Ethics Committee approved this project under the number 90291518.4.0000.5426.

### 2.4. Circular Dichroism Spectroscopy

Circular dichroism spectra were obtained using a Jasco J-815 spectrophotometer (Tokyo, Japan). The wavelength range in which the analyses were recorded ranged from 190 to 260 nm, at a temperature of 25 °C, with three accumulations. To analyze the secondary structure of the CD spectra, analogs were obtained with the peptide at a concentration of 30 μM in aqueous solution (PBS Buffer, pH = 7) and in the well-known structuring solvents trifluoroethanol (TFE—60%), lysophosphatidylcholine micelles (LPC) and sodium dodecyl sulfate (SDS), which were used at concentrations of 10 mM, above the critical micelle concentration (CMC). Furthermore, to analyze the interactions between peptides and bacterial cell components, lipopolysaccharide (LPS) from *E. coli*—Sigma Aldrich^®^ (Sigma-Aldrich, St. Louis, MO, USA) and total lipid extract of *E. coli* (Avanti Polar Lipids, Alabaster, AL, USA ) were added to peptides at concentrations of 1, 5 and 10% at 30 μM in buffer solution (PBS, pH = 7). The spectra of LPS and total lipid extract of *E. coli* were used as negative controls and subtracted from the spectra containing peptides.

### 2.5. Permeability in Different Vesicle Compositions

The permeabilization of lipid vesicles containing carboxyfluorescein was carried out using a spectrofluorometer model Fluorolog-3 FL3-122 from Horiba Jobin Yvon (Newark, NJ, USA) (dimers and tetramers) or in a spectrofluorometer model RF-1501 Shimadzu (Kyoto, Japan) (monomer). For RF-1501, the quantification of the fluorescence intensity was performed every 20 s. Two vesicle compositions were used: (1) 80% 1-palmitoyl-2-oleoyl-sn-glycero-3-phosphoethanolamine (POPE) and 20% 1-palmitoyl-2-oleoyl-sn-glycero- 3-phospho-(10-rac-glycerol) (POPG); and (2) 95% 1-palmitoyl-2-oleoyl-sn-glycero-3-phosphocholine (POPC) and 5% POPG. The percentage of leakage was obtained through the following equation:% release = (F_measured_ − F_initial_)/(F_final_ − F_initial_) × 100,
where F_final_ is the average fluorescence value obtained after the addition of Triton, F_initial_ is the average of the initial values of the run before the addition of the peptide, and F_measured_ are the fluorescence values obtained at each point of the experiment.

### 2.6. PL^pro^ Inhibition

Both the methodology of the enzyme inhibition assay and the cloning, expression and purification of the SARS-CoV-2 PL^pro^ protease were carried out in accordance with the article by Freire et al., 2021 [19]. 

## 3. Results and Discussion

### 3.1. Peptide Synthesis

All p-Bth analogs were synthesized with D-amino or L-amino acids to evaluate the effect of the chirality on their activities (Figure 1). L- and D-monomers (Figure 1A,B) of sequence KKYRYHLKPF were obtained to understand the impact of dimerization on the antibacterial effects and the ability to inhibit the enzymatic activity of PL^pro^ of SARS-CoV-2. The dimer (Figure 1C,D) and tetramer (Figure 1E,F) analogs (L and D isomers) were obtained using the previously described strategy, using one or three Fmoc-Lys(Fmoc)-OH in the C-terminal region [17]. The chains were grown in α- and ε-amino groups, obtaining homo-dimers. Chromatographic profiles and mass spectra are presented in the Appendix A).

### 3.2. Biological Activity

In 2017, the World Health Organization (WHO) published a list of pathogens for which the development of new antimicrobial agents is urgent due to the severe therapeutic limitations. *Enterococcus faecium*, *Staphylococcus aureus*, *Klebsiella pneumoniae*, *Acinetobacter baumannii*, *Pseudomonas aeruginosa,* and *Enterobacter* spp. (ESKAPE) bacteria are microorganisms capable of escaping the action of antibiotics, and are part of the WHO list as high priorities in the development of therapeutic options [1].

Thus, to assess the relevance of the antimicrobial effect of p-Bth analogs, their Minimum Inhibitory Concentration (MIC) and Minimum Bactericidal Concentration (MBC) were analyzed against relevant bacteria. The evaluated strains were the Gram-positives: *S. aureus* ATCC 25923, *S. epidermidis* ATCC 35984, *E. faecalis* ATCC 29212, and *E. faecium* ATCC 70022 (Table 1); and the Gram-negatives: *K. pneumoniae* ATCC 700603, *E. coli* ATCC 25922, *A. baumannii* ATCC 19606, and *P. aeruginosa* ATCC 27853 (Table 2).

MIC values of the peptides are presented in µM and µg/mL. A data analysis of the two measurement units is necessary to adequately compare the biological activity of monomers, dimers, and tetramers with respect to the number of chains. As the monomers have a molecular mass of 1378.67 g/mol, the dimers have a mass of 2868.5 g/mol and the tetramers a mass of 5848.06 g/mol, the results in µg/mL are more correct than those in µM when comparing the molecules, considering the same number of monomeric chains.

Neither the D- nor the L-monomer showed significant antimicrobial activity against Gram-positive (Table 1) and -negative (Table 2) strains at all tested concentrations. These findings agree with Santos-Filho’s hypothesis that the antimicrobial action of the p-BthTX-I peptide relies on its dimeric form, either due to disulfide bonds between cysteine or single or poly-L-lysines [17]. However, the D-monomer showed low bactericidal action against *K. pneumoniae* and *E. coli* at 512 and 256 µg/mL concentrations, respectively.

When comparing the results of the biological activity of the dimer enantiomers against Gram-positive bacterial strains, we observed that the L-dimer showed enhanced antibacterial activity compared to the D-dimer for all tested bacteria. Nevertheless, when comparing the activities of the different dimer enantiomers in Gram-negative bacteria, the D-peptide showed better biological activity than the L-peptide against all bacterial species. Thus, this shows that modifying the dimer’s chirality alters its biological activity in the different tested bacterial species. This finding suggested that the mechanism of action of the peptide p-Bth is different in Gram-negative and Gram-positive bacteria, as found in the peptide (p-BthTX-I)_2_ [16]. Santos-Filho concluded that, in Gram-positive bacteria, (p-BthTX-I)_2_ showed direct action on the bacterial surface, while in Gram-negative bacteria, the peptide was internalized and acted on a specific target. Therefore, the enhanced activity of the D-enantiomeric form in Gram-negative bacteria could be explained by its interactions with specific targets (e.g., proteins and DNA) or by the greater stability of D-enantiomer against proteases [23].

In the case of a peptide analog with four chains, the difference in chirality did not affect the antimicrobial activity once the MIC values were the same among the two enantiomeric tetramer pairs (32 µg/mL or 5 µM) against *K. pneumoniae* and *A. baumannii*.

The increase in the monomeric units of the peptide from two to four did not enhance the antibacterial activity. Exceptions were found for *K. pneumoniae* and *P. aeruginosa* bacteria, in which the tetramers were more active than dimers, suggesting that the inhibitory activity is directed towards a specific target. Peptides presenting the same monomeric chain number have similar activity.

When the MIC values in µM were analyzed, both tetramers showed the same value as the L-dimer against the tested *Staphylococcus*. Furthermore, tetramers were the most potent p-Bth analogs in Gram-negative bacteria, with an MIC value of 5 µM against all strains. These findings showed that, with the same peptide concentration, the tetramerization increases the MIC.

Thus, the D-tetramer was shown to be the most potent analog against both Gram-positive and Gram-negative bacteria. Therefore, the combined strategies of the structural modification of peptides by dendrimerization and chirality changes delivered the best analog in the study.

### 3.3. Hemolysis Assay

Antimicrobial peptides have attracted increasing attention as a promising treatment option. In 2019, approximately 10,000 articles were published on AMPs, with 3000 antimicrobial peptides listed in the Antimicrobial Peptide Database 3 (APD3) [24]. However, AMPs have limitations, and one of them is their toxicity. Thus, hemolytic activity studies were carried out to analyze the potential application of these molecules [25]. The results regarding the hemolytic activity of the p-Bth analogs peptides are shown in Figure 2. Both monomers (Figure 2A,B) and dimeric (Figure 2C,D) peptides showed low hemolytic activity (below 10%) at all evaluated concentrated. The tetramers (Figure 2E,F) showed a slight increase in hemolytic activity compared to the other analogs, with the D-tetramer being more hemolytic.

Then, p-Bth analogs were shown to be extremely promising; they have potent antibacterial activity and a low hemolytic rate at the same concentration. The MIC values of the tetramers ranged from 16 to 64 µg/mL and, at these concentrations, peptides presented 10% hemolysis, similar to those observed for the CAMEL peptide [26]. It is important to emphasize that dendrimerization appears to increase the hemolytic activity of the analogs, as the monomers and dimers maintain a hemolysis percentage below 10%, even at 512 µg/mL, and the D-tetramer reaches 28% hemolysis at this concentration. These results confirmed what was observed with the L-dimer [des-Cys^11^,Lys^12^,Lys^13^-(p-BthTX-I)_2_K] and the eight analogs synthesized through the alanine scanning conducted by Santos-Filho et al. in 2021: a hemolysis rate below 5%, even at a concentration of 512 µg/mL [17]. None of the peptides obtained from the C-terminal region of the homologous PLA_2_ showed hemolytic activity, as this depends on the quaternary structure of the protein [14]. These results indicated that the peptide analogs are promising, with low hemolytic activity and selective toxicity against procaryotes.

### 3.4. Circular Dichroism

The peptide structures are influenced by the environment and directly affect the mechanism of action [27]. The circular dichroism technique is a spectroscopic tool used to determine the secondary structures of peptides [28,29]. Compounds structured in α-helix present a minimum negative at 208 and 216 nm and a positive maximum at 195 nm. β-sheet-structured peptides have a minimum at negative 218 nm and a maximum peak at positive 196 nm. However, proteins and peptides with a random structure have an ellipsis with a minimum peak at 195 nm and a maximum positive peak at 218 nm [30]. CD was also used to investigate interactions between peptides and cellular components, aiming to better understand the mechanism of action of antimicrobial peptides, as such peptides can present structural changes when interacting with cellular components, membrane mimetic environments, and different solutions [30,31,32].

The secondary structure of p-Bth analogs peptides (Figure 3) was obtained at a peptide concentration of 30 μM in an aqueous solution (Buffer PBS, pH = 7.4), with known structuring solvent trifluoroethanol (TFE 60%) and micelles of lysophosphatidylcholine (LPC) and sodium dodecyl sulfate (SDS) at 10 mM, above the critical micelle concentration (CMC), forming the membranes’ mimetic environments.

In an aqueous solution, the L-peptides did not present defined secondary structures. They showed the spectra characteristics of random structures, with negative bands ranging from 195 to 200 nm and a positive band close to 220 [30]. D-peptides in an aqueous solution also showed characteristic spectra of a random structure, with opposite peaks to those observed for L-peptides, bands close to 195–200 nm and a negative band close to 220 nm. This result was expected as the CD of an enantiomeric pair must be the mirror image of the other [33]. In TFE and SDS, the peptide spectra became less characteristic of a random structure, but without bands indicating defined secondary structures. The only exception was the monomer peptide with L-amino acids, which, in SDS, showed a positive maximum at 195 nm and a negative band near 222 nm, characterizing the low content of the α-helix structure [30]. In LPC, all peptides seemed to undergo a slight conformational change, without showing a well-defined structure.

In addition, to evaluate the interaction with cell-wall components in addition to the cell membrane, the CD spectra of the peptides were obtained in the presence of *E. coli* lipopolysaccharide (LPS) Sigma Aldrich^®^ and *E. coli* total lipid extract (Avanti^®^) at concentrations of 1, 5 and 10% (Figure 4 and Figure 5) in relation to the peptide quantity. The interaction between the peptides and cellular components can lead to structural variations that directly impact the mechanism of action of antimicrobial peptides, such as cecropin A and magainin 2, which, in solution, have a random structure, but acquire α-helix secondary structure in the presence of LPS, vesicles or micelles [34].

The L-peptides (Figure 4A,C,E) showed no variation in the CD spectra in the presence of different concentrations of LPS, indicating no interaction with the lipopolysaccharides. However, the D-dimer enantiomer showed a small variation in the CD spectra (Figure 4B,D,F), which suggested that such molecules interact with the component present in the outer membrane of the Gram-negative bacteria. As the D-dimer was more active against Gram-negative bacteria than the L-dimer, this result may be related to the possible interaction between such peptides and the lipopolysaccharide, and the greater specificity of such enantiomers.

In the presence of *E. coli* total lipid extract, as shown in Figure 5, the peptides did not show prominent variations compared to their spectrum in solution, indicating no interaction with lipids.

### 3.5. Vesicle Permeabilization

Peptides that act through mechanisms that lead to membrane damage, such as pore formation (barrel-stave or toroidal) and/or destabilization of the lipid bilayer (carpet-like), are called permeabilizing peptides [35]. The membranes of organisms serve as protective barriers and, therefore, are treated as biological targets for drugs (as antimicrobial peptides) [35]. Other action mechanisms involving metabolic dysfunctions, such as the inhibition of proteins, DNA synthesis and other enzymatic pathways, must also be considered [36].

The vesicle permeabilization assay is used to investigate the mechanism of action of peptides by analyzing the interaction between such molecules and unilamellar vesicles and obtain insights into the sequence–structure–function relationships of membrane-active peptides [35].

Here, we evaluated two systems of membrane mimetics, including vesicles with a composition of 80% POPE and 20% POPG, and another with 95% POPC and 5% POPG. The first is widely used as a model of bacterial membranes, specifically of *E. coli* membrane [37]. Membranes constituting POPC and POPG were used to mimic mammalian membranes due to the presence of choline-containing lipids in animal membranes [14].

Carboxyfluorescein release data in the vesicle of composition 80% POPE and 20% POPG (Figure 6) showed that all analogs, in both enantiomeric forms, have less leakage, with a maximum percentage of 15% reached by the monomers and L-dimer at a concentration of 50 µM. These results showed that the peptides did not have the formation of pores or destabilization of the plasma membrane as their main mechanism of action. Most peptides have an MIC value below 50 µM (Table 1 and Table 2). This observation was made with the peptide p-BthTX-I and (p-BthTX-I)_2_ by Santos-Filho et al. (2015) [14], where the p-BthTX-I and (p-BthTX-I)_2_ peptides did not show fluorophore release in vesicles of the same lipid composition. The results indicated that peptides derived from the homologous PLA_2_ protein BthTX-I and analogs show antimicrobial action that is promoted by the action on specific targets, either on the membrane or in the intracellular medium. To confirm this hypothesis, more complex and elucidative studies are needed.

The leakage of CF in 95% POPC and 5% POPG vesicles (Figure 7) was greater than that in 80% POPE/20% POPG, but did not exceed the value of 40%. This indicated that dimerization or/and dendrimerization using poly-L-lysine increases the interaction between peptides and vesicles. The monomers led to low leakage values, while the dimers and tetramers significantly increased permeabilization [14]. However, this interaction is small compared to other peptides, such as the W^6^-Hy-a1 peptide studied by Crusca Jr (2011) [38], which, at a concentration of 4 µM, showed about 90% leakage in the studied LUVs. The D-dimer analog at 50 µM is less active than the peptide used in 4 µM, and only 20% of CF leakage was found.

The D-peptides in 95% POPC and 5% POPG vesicles showed 15, 20, and 35% fluorescein leakages for the monomer, dimer, and tetramer, respectively. These results agree with those obtained in the hemolysis assay of the peptides, where the most hemolytic peptide was the tetramer analog. The red blood cell contains a higher quantity of the POPC in its membrane. Gogoi et al. (2021) also showed that the hemolytic activity increased with the branching of the peptide [39].

In addition, dimeric peptides showed a concentration-dependent relationship with CF leakage; at 50 µM, the observed leakage was greater than that at 10 µM. These results could indicate a hybrid leakage model [35], where most permeabilization of the fluorescein occurs immediately after the addition of peptides to vesicles (burst of leakage), followed by a dramatic slowing of leakage over a short period of time. This slight fluorescein leakage could indicate that peptides may induce small and transient apertures in the target membrane, as observed with magainin 2, which forms pores that are not as stable, but are transient structures [40]. When the transient pores are closed, the peptide chains translocate across the bilayer and could be adsorbed in the head group region in the cell or act on other action targets, such as enzymes and DNA.

This hypothesis is confirmed by antimicrobial data, where dendrimeric analogs show small leakage in the membrane model but potent antimicrobial activity.

### 3.6. Inhibition of SARS-CoV-2 PL^pro^ Enzymatic Activity

In December 2019, the first case of viral infection with a new coronavirus was reported in China, which was later named severe acute respiratory syndrome coronavirus 2 (SARS-CoV-2), and the disease caused by this was named coronavirus-2019 or COVID-19 by the WHO. This disease has taken on great proportions, causing a pandemic scenario and great global concern. Even with the improvements in disease control, there is still a need for new, specific antiviral therapeutic options [41]. As antimicrobial peptides also have antiviral action [18], the L-dimer, D-dimer and L-tetramer, in addition to other p-Bth analogs obtained through alanine scanning studies, were evaluated regardinng their ability to inhibit SARS-CoV-2 infection, showing percentage inhibitions of infection of 27, 69 and 54%, respectively [19]. These data were published in a previous study that showed that modifications of chirality more than doubled the dimer’s ability to inhibit SARS-CoV-2 infection, and the tetramer led to significantly improved inhibition activity. The antiviral activity of p-Bth analogs was attributed to the peptide’s ability to inhibit the enzymatic activity of papain-like cysteine protease (PL^pro^), which is important in the cleavage and processing of polyproteins that are part of the viral replication complex and affects the innate immune responses. Thus, by inhibiting PL^pro^, the peptides derived from p-BthTX-I promote dysfunction in the replication and viral propagation of SARS-CoV-2.

Aiming to understand the need for the dimeric or tetrameric forms of the peptide in the enzymatic inhibition of the papain-like cysteine protease, we assessed the enzymatic inhibition of PL^pro^ by the L- and D-monomers. Moreover, to obtain better molecules by joining the dendrimerization and D-compounds in the same molecule, we also assessed the enzymatic inhibition promoted by the D-tetramer.

The inhibition of PL^pro^ of SARS-CoV-2 at 10 μM of the p-BthTX-I analog peptides was maintained when the peptides were in monomer form, showing only a slight reduction, which was more significant for the D-monomer (Table 3). The IC_50_ values of the peptide analogs showed comparable inhibition values (Table 3) and profiles. This is a strong indication that the antiviral activity of the peptide did not depend on its dimeric or tetrameric form. These findings contrast with the observed antibacterial activity, which showed a drastic reduction in potency when the analogs occurred in the form of monomers. This phenomenon can be explained by the vast differences between the structures, components and physiology of bacteria and viruses and, consequently, the different targets in which the peptides perform their antibacterial and antiviral action. A preliminary structure–activity relationship (SAR), which modeled the peptide binding to PL^pro^ from SARS-CoV-2, suggested that the L-dimer is in close contact with key amino acids involved in the enzymatic catalysis [19].

Differing from what was expected, the combination of the two structural modification strategies in the D-tetramer did not lead to an increase in the percentage of inhibition of the enzymatic activity of PL^pro^. However, all the peptides in the work, except for the D-monomer, led to PL^pro^ inhibition percentages close to 100% and were interesting molecules, whose antiviral actions against SARS-CoV-2 should be studied in more depth.

## 4. Conclusions

The evaluation of the antibacterial activity of such molecules led to the confirmation of the hypothesis that the antimicrobial action of p-Bth analog peptides relies on their dimeric or dendrimeric form.

By contrast, the inhibition of the enzymatic activity of the PL^pro^ of SARS-CoV-2 was maintained when the peptides were in monomer form. Therefore, the studied peptides led to interesting inhibition percentages of the enzymatic activity of PL^pro^ (85 to 98% at 10 µM), and stand out as molecules whose antiviral actions against SARS-CoV-2 should be studied in more depth.

The peptides’ p-Bth analogs had high antibacterial potency and an increased ability to inhibit the PL^pro^ enzyme of SARS-CoV-2, proving to be promising for the development of new drugs.

## Figures and Tables

**Figure 1 pharmaceutics-15-00436-f001:**
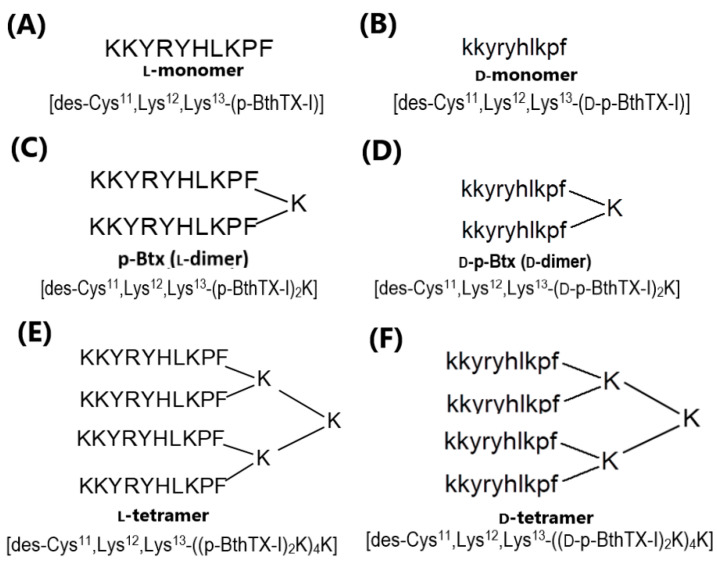
p-Bth peptide analogs, structure, codes and sequences. (**A**) L-monomer, (**B**) D-monomer, (**C**) L-dimer, (**D**) D-dimer, (**E**) L-tetramer, and (**F**) D-tetramer.

**Figure 2 pharmaceutics-15-00436-f002:**
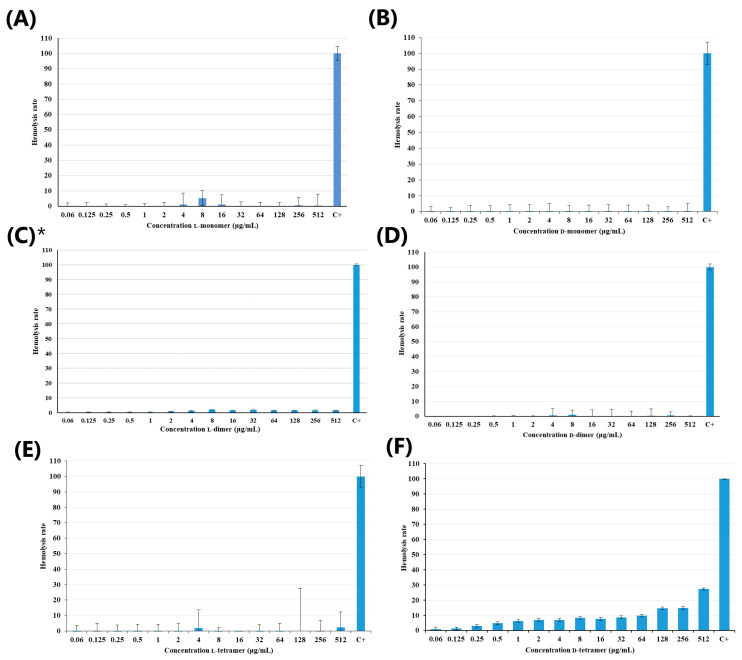
Hemolytic activity of peptides: (**A**) L-monomer, (**B**) D-monomer, (**C**) L-dimer, (**D**) D-dimer, (**E**) L-tetramer, and (**F**) D-tetramer at hemolytic rate (%) as a function of the peptide concentration µg/mL. Results provided in the article by Santos-Filho et al., 2021 [17].

**Figure 3 pharmaceutics-15-00436-f003:**
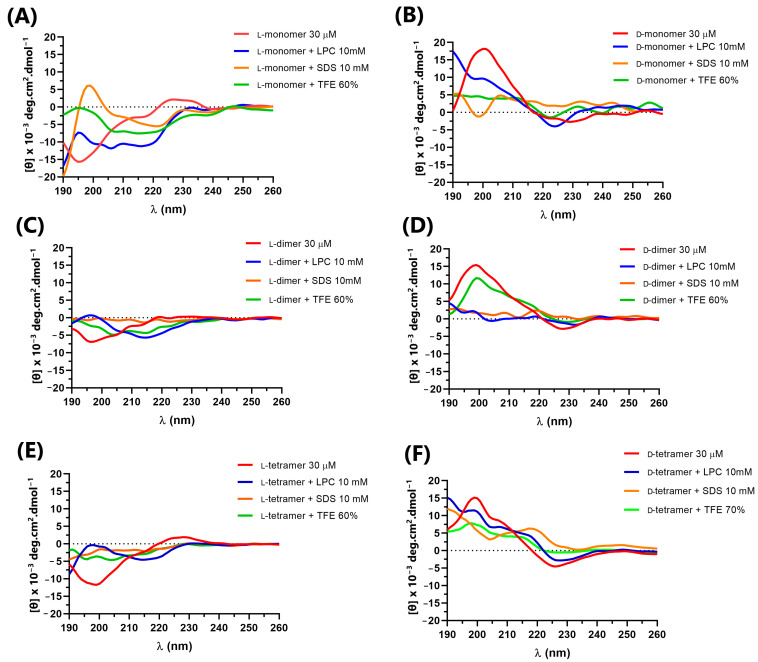
Circular dichroism of peptides: (**A**) L-monomer, (**B**) D-monomer, (**C**) L-dimer, (**D**) D-dimer, (**E**) L-tetramer, and (**F**) D-tetramer at 30 µM in PBS, LPC, SDS and TFE.

**Figure 4 pharmaceutics-15-00436-f004:**
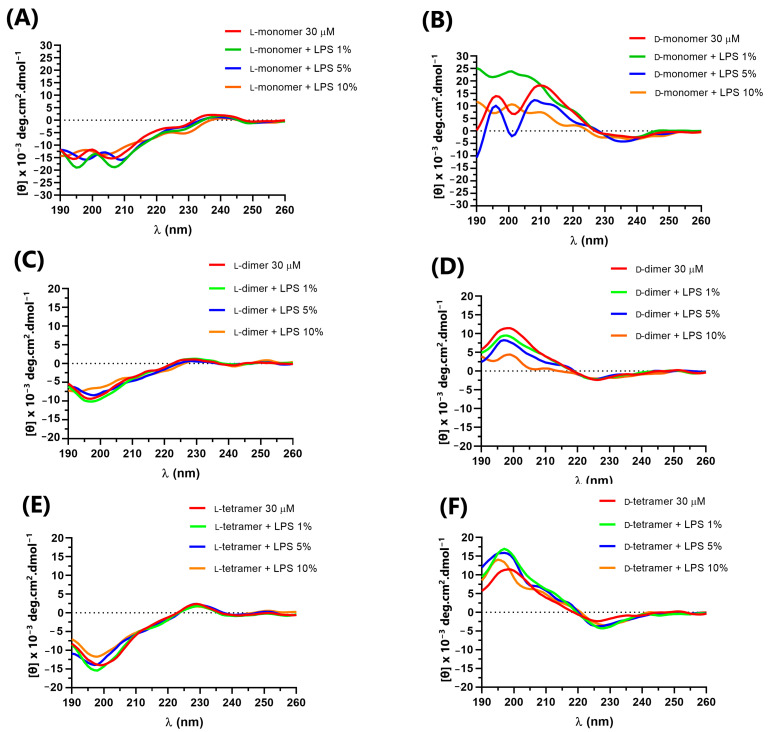
Circular dichroism of peptides: (**A**) L-monomer, (**B**) D-monomer, (**C**) L-dimer, (**D**) D-dimer, (**E**) L-tetramer, and (**F**) D-tetramer at 30 µM in PBS and in the presence of 1, 5 and 10% LPS.

**Figure 5 pharmaceutics-15-00436-f005:**
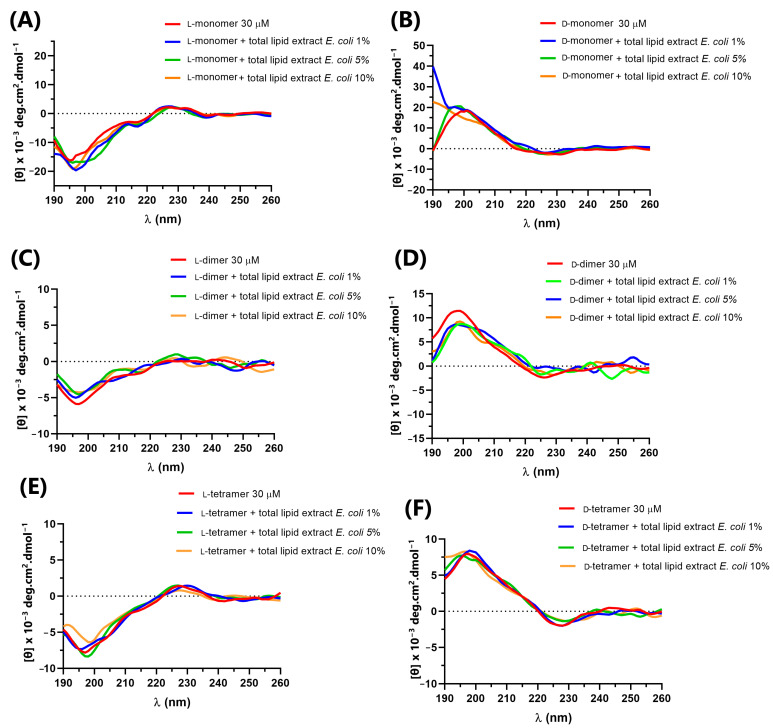
Circular dichroism of peptides: (**A**) L-monomer, (**B**) D-monomer, (**C**) L-dimer, (**D**) D-dimer, (**E**) L-tetramer, and (**F**) D-tetramer at 30 µM in PBS and in the presence of 1, 5 and 10% *E. coli* total lipid extract.

**Figure 6 pharmaceutics-15-00436-f006:**
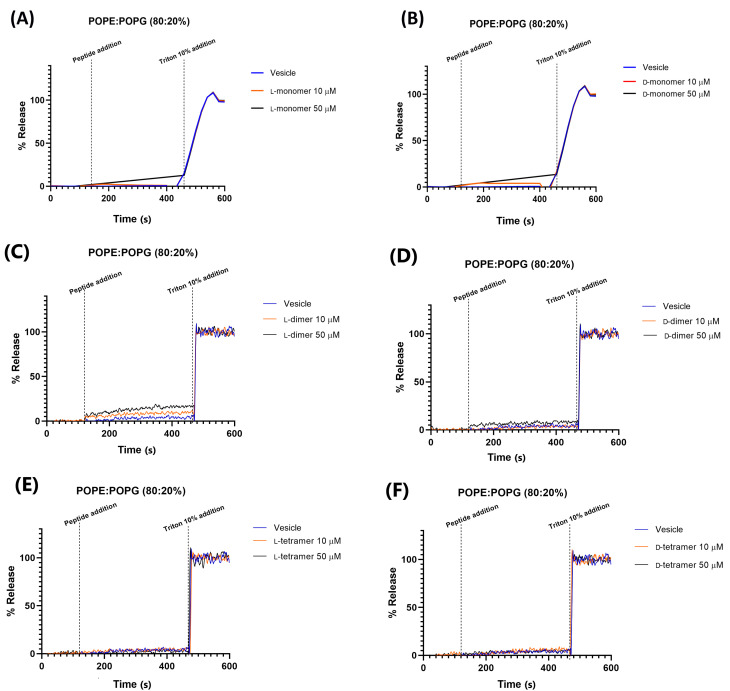
Permeabilization in 80% POPE and 20% POPG vesicles of the peptides: (**A**) L-monomer, (**B**) D-monomer, (**C**) L-dimer, (**D**) D-dimer, (**E**) L-tetramer, and (**F**) D-tetramer at concentrations of 10 and 50 µM.

**Figure 7 pharmaceutics-15-00436-f007:**
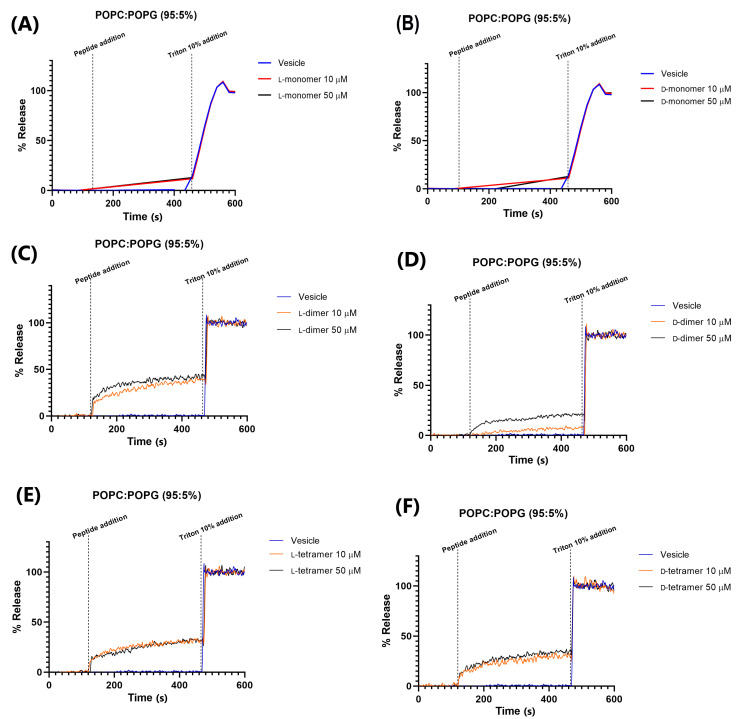
Permeabilization in 95% POPC and 5% POPG vesicles of the peptides: (**A**) L-monomer, (**B**) D-monomer, (**C**) L-dimer, (**D**) D-dimer, (**E**) L-tetramer, and (**F**) D-tetramer at concentrations of 10 and 50 µM.

**Table 1 pharmaceutics-15-00436-t001:** Minimal inhibitory, bactericidal concentration and antibacterial activity of p-BthTX-I analogs against Gram-positive bacteria.

Peptides	*S. aureus* ATCC 25923				*S. epidermidis* ATCC 35984				*E. faecalis* ATCC 29212				*E. faecium* ATCC 700221			
	MIC μg/mL	MIC µM	MBC μg/mL	Act	MIC μg/mL	MIC µM	MBC μg/mL	Act	MIC μg/mL	MIC µM	MBC μg/mL	Act	MIC μg/mL	MIC µM	MBC μg/mL	Act
**L** **-monomer**	>512	>371	>512	ND	>512	>371	>512	ND	>512	>371	>512	ND	>512	>371	>512	ND
**D** **-monomer**	>512	>371	>512	ND	>512	>371	>512	ND	>512	>371	>512	ND	>512	>371	>512	ND
**L** **-dimer**	32	11	128	Bc	8	2	16	Bc	64	22	128	Bc	16	5	>64	Bs
**D** **-dimer**	256	89	>512	ND	16	5	16	Bc	512	178	512	Bc	64	22	>256	Bs
**L** **-tetramer**	64	10	256	Bc	16	2	32	Bc	64	10	128	Bc	16	2	32	Bc
**D** **-tetramer**	64	10	64	Bc	16	2	16	Bc	32	5	32	Bc	16	2	16	Bc

Abbreviations: MIC, minimum inhibitory concentration; MBC, minimum bactericidal concentration; Act, antibacterial activity; ND, not determined; Bc, bactericidal, Bs, bacteriostatic.

**Table 2 pharmaceutics-15-00436-t002:** Minimal inhibitory, bactericidal concentration and antibacterial activity of p-BthTX-I analogs against Gram-negative bacteria.

Peptides	*K. pneumoniae* ATCC 700603				*E. coli * ATCC 25922				*A. baumannii* ATCC 19606				*P. aeruginosa* ATCC 27853			
	MIC μg/mL	MIC µM	MBC μg/mL	Act	MIC μg/mL	MIC µM	MBC μg/mL	Act	MIC μg/mL	MIC µM	MBC μg/mL	Act	MIC μg/mL	MIC µM	MBC μg/mL	Act
L-monomer	>512	>371	>512	ND	>512	>371	>512	ND	>512	>371	>512	ND	>512	>371	>512	ND
D-monomer	512	371	512	Bc	256	185	256	Bc	>512	>371	>512	ND	>512	>371	>512	ND
L-dimer	>128	>44	ND	ND	32	11	64	Bc	256	89	>512	ND	>512	>178	ND	ND
D-dimer	64	22	128	Bc	32	11	32	Bc	16	5	64	Bc	256	89	512	Bc
L-tetramer	32	5	>128	Bs	32	5	>128	Bs	32	5	64	Bc	32	5	64	Bc
D-tetramer	32	5	32	Bc	32	5	>128	Bs	32	5	32	Bc	32	5	64	Bc

Abbreviations: MIC, minimum inhibitory concentration; MBC, minimum bactericidal concentration; Act, antibacterial activity; ND, not determined; Bc, bactericidal, Bs, bacteriostatic.

**Table 3 pharmaceutics-15-00436-t003:** PL^pro^ activity by peptide analogs of p-BThTX-I at the concentration of 10 μM and IC_50_ values. Representative concentration–response inhibition curves against PL^pro^ from SARS-CoV-2 are presented in the Appendix A).

Peptide	Inhibition (%) at 10 µM	IC_50_ μM
L-monomer	93.30 ± 0.00	2.1 ± 0.1
D-monomer	85.00 ± 3.00	3.0 ± 1
L-dimer *	98.05 ± 1.62	2.4 ± 0.1
D-dimer *	96.35 ± 0.91	1.30 ± 0.03
L-tetramer *	98.40 ± 0.00	1.40 ± 0.02
D-tetramer	94.00 ± 3.00	2.7 ± 0.5

* Results provided in the article by Freire et al., 2021 [19].

## Data Availability

Not applicable.

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
