# Peer review of "Effects of Dimerization, Dendrimerization, and Chirality in p-BthTX-I Peptide Analogs on the Antibacterial Activity and Enzymatic Inhibition of the SARS-CoV-2 PLpro Protein"

_pharmaceutics, 2023, doi:10.3390/pharmaceutics15020436_

Round 1
Reviewer 1 Report
In the current manuscript, the authors extended their previous research by performing dimerization and tetramerization of formerly known p-Bth peptide inhibitors. The synthesis was achieved by immobilizing the Fmoc-Lys(Fmoc)-OH on the solid support followed by sequential couplings. This is a smart strategy to combine the multiple monomers to further enhance biological efficiency. The synthesized peptide derivatives were characterized by HPLC and the LC-MS and their secondary structure was confirmed by CD spectra. The authors further demonstrated the effect of these multi-monomers by checking their anti-microbial activity against gram-positive and -negative bacteria, and anti-viral activity against SARS-CoV-2 PLpro protein. The authors also performed the hemolytic assay to check the toxicities and the vesicle permeabilities of synthesized peptide derivatives.
Overall, the current report indicates/proves that the peptide dendrimerization and chirality play key roles in determining the biological efficiencies as the D-tetramer was found to be better in this case.
Minor comments:
1. Please elaborate on the AMPs in the introduction part of the manuscript.
2. Did the authors check the PLpro activity of peptide derivates at low concentrations in Table III? Please include those details in the main text.
3. It would be beneficial for the readers if the authors discuss the mechanistic aspects of these peptide derivatives.
4. Can authors provide the complete protocol for peptide synthesis including the yields in the supporting information file?
5. Can authors include the deconvoluted masses in the supporting information file?
Author Response
Reviewer #1:
In the current manuscript, the authors extended their previous research by performing dimerization and tetramerization of formerly known p-Bth peptide inhibitors. The synthesis was achieved by immobilizing the Fmoc-Lys(Fmoc)-OH on the solid support followed by sequential couplings. This is a smart strategy to combine the multiple monomers to further enhance biological efficiency. The synthesized peptide derivatives were characterized by HPLC and the LC-MS and their secondary structure was confirmed by CD spectra. The authors further demonstrated the effect of these multi-monomers by checking their anti-microbial activity against gram-positive and -negative bacteria, and anti-viral activity against SARS-CoV-2 PLpro protein. The authors also performed the hemolytic assay to check the toxicities and the vesicle permeabilities of synthesized peptide derivatives.
Overall, the current report indicates/proves that the peptide dendrimerization and chirality play key roles in determining the biological efficiencies as the D-tetramer was found to be better in this case.
Minor comments:
- Please elaborate on the AMPs in the introduction part of the manuscript.
A.: We would like to thank referee for this comment. It is done.
- Did the authors check the PLproactivity of peptide derivates at low concentrations in Table III? Please include those details in the main text.
A.: Yes, the IC50 was added in Table II and the concentration–response inhibition curves against PLpro from SARS‐CoV‐2 for L-monomer (A), D-monomer and D-tetramer was added in figure S7.
- It would be beneficial for the readers if the authors discuss the mechanistic aspects of these peptide derivatives.
A.: We would like to thank referee for this comment. Really a better discussion was necessary to clarify its question. One paragraph was added about the action mechanism of the peptide in PLpro enzyme.
- Can authors provide the complete protocol for peptide synthesis including the yields in the supporting information file?
A.: Additional information about the protocol for peptide synthesis was added in supporting information file.
- Can authors include the deconvoluted masses in the supporting information file?
A.: Thank you the reviewer for pointing this out. The molecular weight of the peptides and the experimental values were placed in Table S1.

Reviewer 2 Report
The paper "Effects of dimerization, dendrimerization, and chirality in p-BthTX-I peptide analogs on the antibacterial activity and enzymatic inhibition of the SARS-CoV-2 PLpro protein" is interesting and made with adequate criteria.The study is well designed but the manuscript must be significantly improved.
Detailed review report:
The lines are not numbered, which makes the reviewer's work difficult.
1. There are typos in the text.
2. Figure 1 and table 3 are too big, please make its smaller.
3. Table I should be in section 3 (Results), after table I.
4. In addition, tables I and II are completely illegible, which makes it impossible to evaluate the results.
In Tab. I and II (after correction) standard deviations need be introduced.
Author Response
The paper "Effects of dimerization, dendrimerization, and chirality in p-BthTX-I peptide analogs on the antibacterial activity and enzymatic inhibition of the SARS-CoV-2 PLpro protein" is interesting and made with adequate criteria. The study is well designed but the manuscript must be significantly improved.
Detailed review report:
The lines are not numbered, which makes the reviewer's work difficult.
- There are typos in the text.
A.: We would like to thank referee for this comment. The article was revised to ensure that the English language is clear and free of errors.
- Figure and table 3 are too big, please make its smaller.
A.: Thank you the reviewer for pointing this out. It is fixed.
- Table I should be in section 3 (Results), after table I.
A.: Thank you the reviewer for pointing this out. It is fixed.
- In addition, tables I and II are completely illegible, which makes it impossible to evaluate the results.
A.: Thank you the reviewer for pointing this out. It is fixed.
- In Tab. I and II (after correction) standard deviations need be introduced.
A.: We thank for this comment. The mode was used because the mean, in this case, could describe a value that is not necessarily inhibitory. So, the standard deviation does not apply in this case.

Round 2
Reviewer 2 Report
I recommend the article for publication in present form.